# Air pollution control strategies directly limiting national health damages in the US

Yang Ou [1,2,5], J. Jason West [1], Steven J. Smith [3], Christopher G. Nolte [4] & Daniel H. Loughlin [4 ✉]

Exposure to fine particulate matter ($PM_{2.5}$) from fuel combustion significantly contributes to global and US mortality. Traditional control strategies typically reduce emissions for specific air pollutants and sectors to maintain pollutant concentrations below standards. Here we directly set national $PM_{2.5}$ mortality cost reduction targets within a global human-earth system model with US state-level energy systems, in scenarios to 2050, to identify endogenously the control actions, sectors, and locations that most cost-effectively reduce $PM_{2.5}$ mortality. We show that substantial health benefits can be cost-effectively achieved by electrifying sources with high primary $PM_{2.5}$ emission intensities, including industrial coal, building biomass, and industrial liquids. More stringent $PM_{2.5}$ reduction targets expedite the phaseout of high emission intensity sources, leading to larger declines in major pollutant emissions, but very limited co-benefits in reducing $CO_2$ emissions. Control strategies limiting health damages achieve the greatest emission reductions in the East North Central and Middle Atlantic states.

---

[1] Department of Environmental Sciences and Engineering, University of North Carolina at Chapel Hill, Chapel Hill, NC 27599, USA. [2] ORISE Participant at the U.S. Environmental Protection Agency, 109 T.W. Alexander Drive, Research Triangle Park, NC 27711, USA. [3] Joint Global Change Research Institute, Pacific Northwest National Laboratory, 5825 University Research Court, College Park, MD 20740, USA. [4] Center for Environmental Measurement and Modeling, U.S. Environmental Protection Agency, 109 T.W. Alexander Drive, Research Triangle Park, NC 27711, USA. [5] Present address: Joint Global Change Research Institute, Pacific Northwest National Laboratory, 5825 University Research Court, College Park, MD 20740, USA. ✉email: Loughlin.Dan@epa.gov

A ir pollution control in the USA is focused on compliance with emission limits and meeting air quality standards. Over recent decades, US air quality regulations have greatly reduced emissions from traditional polluting sources, such as sulfur dioxide ($SO_2$) and nitrogen oxides ($NO_x$) from coal-fired power plants and on-road transportation. These pollutants, as well as directly emitted $PM_{2.5}$, contribute to ambient $PM_{2.5}$ levels. As a result of these emission reductions, the national average annual $PM_{2.5}$ concentration has declined by 42% from 2000 to 2015[1], and the corresponding $PM_{2.5}$ mortality burden has also decreased significantly[2,3]. Despite decreasing $PM_{2.5}$ concentrations, recent epidemiological evidence indicates there are still significant adverse health effects associated with $PM_{2.5}$ exposure at approximately current US concentration levels[4,5].

A primary goal of air pollution control is to protect human health. Air pollution health risks can be most cost-effectively addressed through strategies that simultaneously consider multiple pollutants from multiple source sectors, as well as factors such as long-term population and economic growth. Analyses of the impacts of proposed regulations[6–9] typically follow an emission-impact paradigm, in which a first step is to identify emission-reduction targets that meet air pollutant concentration standards. Next, control strategies are devised to achieve the emission reduction targets. Candidate strategies are then evaluated using air quality and health impact modeling to estimate the benefits of the proposed actions. However, since the emission-reduction target determination and air quality and health modeling are decoupled, the complexity and time requirements of these models limit the ability to explore more than a handful of control strategies. In addition, there is no guarantee that the proposed control strategy or the specific combination of control measures is the most cost-effective pathway. Alternatively, air quality management studies[10–17] have explored the development of multi-pollutant control strategies using an optimization framework. While these studies in general confirm that integrated planning across multiple pollutants is more cost-effective than individual mitigation actions, their focus generally has been on a relatively small spatial scale[12,13,15,16] or small number of pollutants[14] over a short period[10,12,13,17], due to the complexity of their optimization algorithms required to address exposure-outcome nonlinearties.

To our knowledge, this is the first study to model the most cost-effective actions to directly achieve a specified reduction in national health damages of air pollution. In contrast with traditional strategies based on absolute or relative emission reductions for specific air pollutants or sectors, this paper focuses on $PM_{2.5}$-related mortality costs (hereinafter referred to as PMMC), which account for over 90% of the total monetized $PM_{2.5}$ health impact[18]. We investigate which sectors, technologies, and states can most cost-effectively reduce emissions to achieve various levels of PMMC reduction targets. This is accomplished by integrating a representation of the monetized $PM_{2.5}$ attributable mortality of inorganic air pollutant emissions[19,20] into a version of the Global Change Assessment Model with state-level representation of the US energy system (GCAM-USA)[21,22], which provides a platform that allows a national mortality reduction target to be applied. PMMC is modeled in GCAM-USA by multiplying a pollutant-, source category-, and state-specific PMMC coefficient ($ \text{Tg}^{-1}$) with the corresponding pollutant emissions (Tg) for each technology. Within this framework, we impose PMMC reduction targets, and GCAM-USA finds how to meet those targets cost-effectively while meeting energy demands.

Our results suggest that even under a scenario with limited transitions from coal toward renewable energy, substantial health benefits can still be achieved cost-effectively by using electricity to replace sources with high primary $PM_{2.5}$ emission intensities,

including industrial coal, building biomass, and industrial liquids. This finding also holds under a scenario with a more aggressive transition toward renewable energy.

## Results

**Reference scenarios.** We define a reference scenario (BASE REF) that includes updated baseline assumptions about technology trajectories harmonized with the 2018 Energy Information Administration (EIA) Annual Energy Outlook (AEO)[23], as well as an alternative reference scenario (HR REF) with much higher shares of wind and solar generation (e.g., 48% versus 25% in 2050), intended to reflect a continuation of recent, rapid growth in renewables. For both, major air pollutant and $CO_2$ emission regulations currently in place are represented[24], assuming no additional future regulations ("Methods"). Electric sector coal use in HR REF decreases by 51% from 2015 to 2050, compared with the 8% increase in BASE REF (Supplementary Fig. 1). Projected emissions of major air pollutants are also considerably lower in HR REF (Supplementary Fig. 2). In 2050, $SO_2$ emissions decrease by 47% and 21% from 2015 in HR REF and BASE REF, correspondingly. Primary $PM_{2.5}$ emissions decrease by 65% and 27% from 2015 in HR REF and BASE REF, correspondingly. These additional pollutant emission reductions lead to a much lower baseline PMMC projection.

Relative to BASE REF, five additional scenarios (US10–US50) apply increasingly stringent national PMMC reduction targets of 10% (US10) to 50% (US50) reductions in 2050 relative to BASE REF in 2050. Similarly, a 50% PMMC reduction target is applied to HR REF (HR US50). We focus on the BASE scenarios in this study because the coal and renewable assumptions under BASE REF pose greater challenges for reducing PMMC ("Methods"). For simplicity, REF and US10–US50 throughout the rest of the study refer to scenarios based on BASE REF. HR REF and HR US50 are used to evaluate the robustness of the major findings using BASE.

Under REF, PMMC in the US declines in the near-term, but then starts to level off around 2035 (Supplementary Fig. 3). This decline is driven by both socioeconomic and energy factors[21]. Decreased energy intensity (energy use per unit GDP), more stringent emission controls, and decreased baseline mortality rates contribute to lower PMMC. Countervailing factors, such as growing population, higher income levels (which increase willingness-to-pay), and higher future energy demands, contribute to the leveling off of PMMC after 2035.

**Effects of 50% PMMC reductions.** For both BASE and HR scenarios, reducing PMMC by 50% leads to very little change in national total energy use in 2050, relative to REF (Fig. 1a, b). The PMMC reduction in US50 is achieved mainly by substitution of industrial coal, building biomass, and industrial liquids with electricity (Fig. 1b). Compared with REF, electric sector energy input for US50 is 4.2% higher in 2050, primarily from natural gas and clean energy, such as wind and solar. Total energy use in the industry and building sectors decreases 5.2% and 1.1% on an end-use equivalent basis, respectively, while PMMC from these sectors is reduced by 71% and 65%, respectively (Fig. 1e). The decrease in industrial coal, building biomass, and industrial liquids in US50 is equivalent to only 2% of the total energy consumption in 2050, but contributes 87% of the total PMMC reductions relative to REF. Here, we call these high-emission-intensity sources because of their disproportionately high contributions to PMMC.

Under the HR US50 (Fig. 1c, f), the decrease in these high-emission-intensity sources is also equivalent to 2% of the total energy consumption in 2050 but still contributes to 70% of the PMMC redutions, with additional contributions from decreased

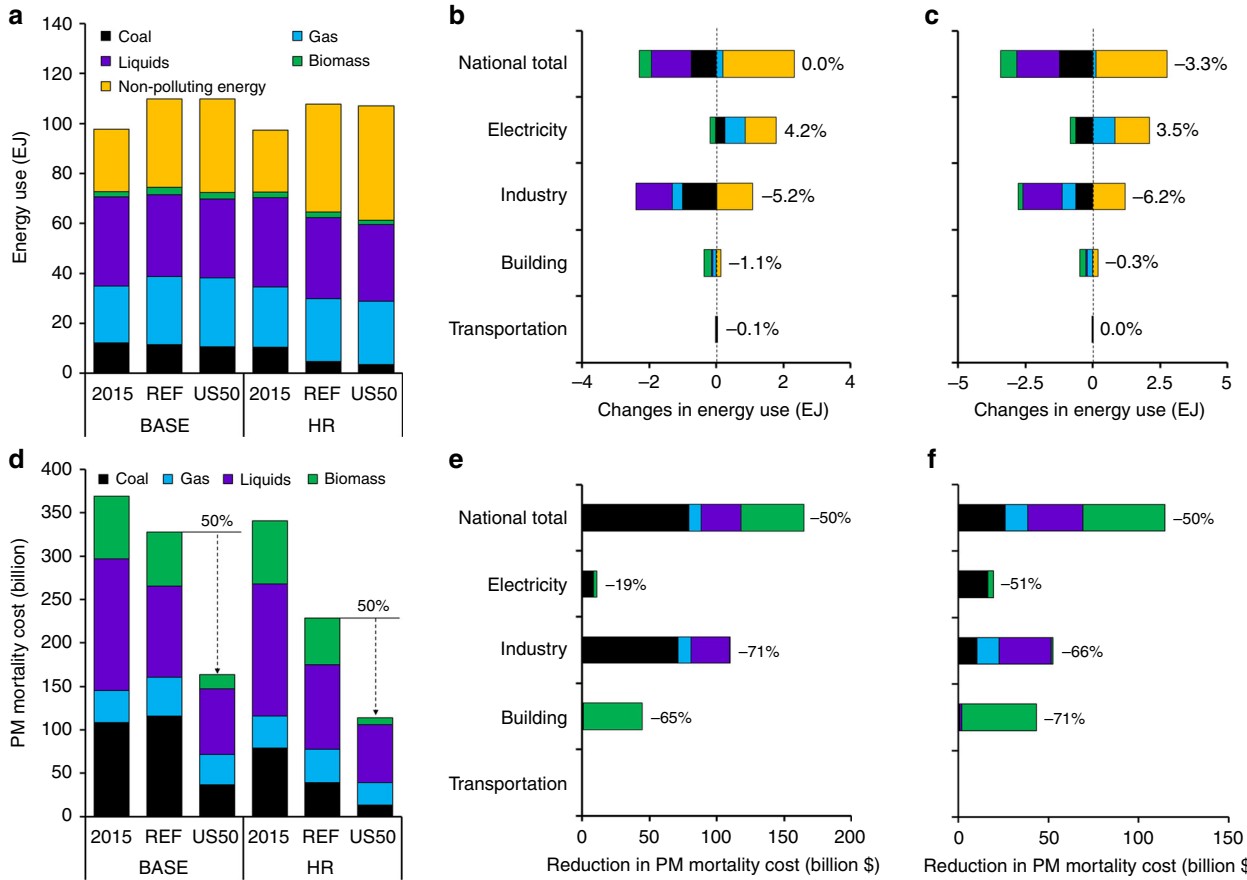

**Fig. 1 National energy system and PM$_{2.5}$ mortality cost responses to the 50% PM$_{2.5}$ mortality cost reduction (US50) based on two reference scenarios.** National total energy use (EJ) by fuel (**a**), sectoral changes in energy use (EJ) by fuel for BASE US50 relative to BASE REF (**b**) and HR US50 relative to HR REF (**c**); national total PM$_{2.5}$ mortality costs (billion 2018$) by fuel (**d**), and sectoral changes in PM$_{2.5}$ mortality costs (billion $) by fuel for US50 reductions under BASE (**e**), and HR scenarios (**f**). With the exception of the bars labeled 2015, all values are for 2050. Changes in the transportation sector are negligible. The nonpolluting energy category includes geothermal, hydropower, wind, and solar power in the electricity sector, and hydrogen and electricity in end-use sectors. Source data are provided as a Source Data file.

electric coal use. Electric coal use is further reduced for the following two reasons. First, wind and solar costs are much lower in HR US50 than BASE US50, making it more cost-effective to displace coal generation. Second, the HR US50 mortality reduction target, which is relative to the HR 2050 reference case value, is more stringent than the BASE US50 target because the HR REF already has a lower baseline PMMC compared to BASE REF (Fig. 1).

Overall and sectoral reductions of PMMC in US50 are dominated by reductions in primary PM$_{2.5}$ emissions (Supplementary Fig. 2). Electricity generation is not a major sector for achieving PMMC reductions in these scenarios, because existing regulations incorporated in REF already reduce NO$_x$, SO$_2$, and primary PM$_{2.5}$ emissions from the electric power sector by 2050, particularly from coal-fired power plants[24,25]. US50 also has a negligible effect on the transportation sector, because emission standards for new on-road vehicles result in a much lower polluting fleet by 2050. In addition, these modeling results suggest that large-scale vehicle electrification is less cost-competitive for reducing PM$_{2.5}$ compared with actions available in other sectors under the assumptions in REF, which considers recent market trends of battery vehicle costs (Supplementary Table 1) and fuel economy requirements (Supplementary Table 2; "Methods").

This synergy of clean power and electrified end-use technologies is the key to achieve substantial multisector PMMC reductions. In US50, 86% of the additional electric sector energy

input in 2050 relative to REF is through nonpolluting energy sources (such as wind and solar) or gas. Compared with REF, electricity generation from wind and solar is 15% (0.8 EJ) larger in 2050, while generation from oil and biomass decreases 18% (0.06 EJ) (Supplementary Table 3). This structural change in the electric sector lowers pollutant emissions. For end-use sectors, replacing coal, liquids, and gas with electricity achieves significant emission reductions for industrial SO$_2$ (−62%) and NO$_x$ (−34%), accounting for over 90% of the total reductions in SO$_2$ and NO$_x$ in US50 in 2050 (Supplementary Fig. 2). Electrified end-use technologies can also promote energy efficiency. For example, electric heat pumps for residential heating are cleaner and more efficient than traditional combustion-based technologies for space heating, which explains the slightly reduced building energy use in US50.

**Effects of PMMC reduction stringency.** The choice of which pollutants to reduce from which sectors is determined endogenously by both their respective dollar-per-ton mortality impact, the cost of control options, and alternative technologies. Compared with REF, US50 leads to 51% and 32% reductions of the primary PM$_{2.5}$ and SO$_2$ emissions in 2050, respectively, but reduces NO$_x$ emissions by only 11% (Fig. 2). The primary PM$_{2.5}$ and SO$_2$ reductions in 2030 and 2035 under US50 are already close to the corresponding reductions achieved by REF in 2050. Further

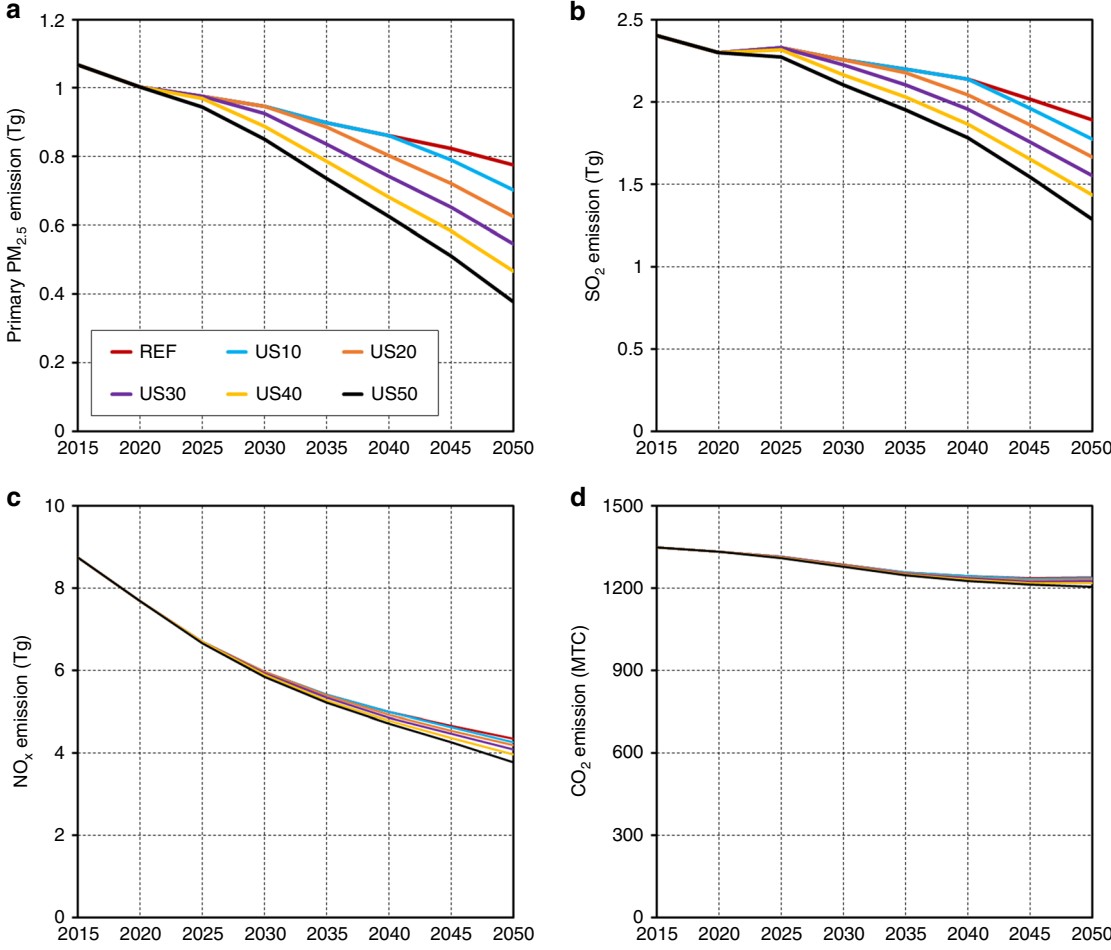

**Fig. 2 Effects of PM$_{2.5}$ reduction stringency on national emissions.** Compared with REF, more stringent PM$_{2.5}$-related mortality costs (PMMC) reduction targets lead to greater reductions of the primary PM$_{2.5}$ (**a**) and SO$_2$ (**b**) emissions in 2050, respectively, but minor reductions of NO$_x$ (**c**) and CO$_2$ emissions (**d**). National emissions include anthropogenic sources from electricity, industry, transportation, and buildings sectors in continental US. Emissions from AK and HI are not included. Source data are provided as a Source Data file.

decreasing emissions of primary PM$_{2.5}$ and SO$_2$ more cost-effectively achieves PMMC reductions than decreasing NO$_x$ emissions for two reasons. First, NO$_x$ has the lowest \$ ton$^{-1}$ PMMC coefficient (Supplementary Table 4), making NO$_x$ reduction less cost-effective as a means of reducing PMMC. Second, remaining PM$_{2.5}$ and SO$_2$ emissions are largely from the same high-emission-intensity sources (Supplementary Figs. 4–6) that can be reduced simultaneously and cost-effectively by technology switching. In contrast, the largest remaining source of NO$_x$ emissions is liquid fuels in the transportation sector, which is more expensive to replace.

While CO$_2$ reductions are a co-benefit from reducing PMMC (Fig. 2d), this co-benefit is minor because the high-emission-intensity sources affected by PMMC reduction targets are minor contributors to energy consumption in the industry and building sectors (Supplementary Figs. 7 and 8), as well as relatively small sources of CO$_2$. Thus, while many studies have found substantial air quality co-benefits of CO$_2$ reductions[22,26,27], our findings suggest that emission reductions driven solely by air pollution health improvement targets do not necessarily result in large CO$_2$ emission reductions under the current US energy structure (see "Discussion"). This finding is valid under both BASE and HR energy trajectories (Supplementary Fig. 9).

More stringent PMMC reductions promote more rapid electrification and technology replacement as early as 2020, which is the first modeling period with PMMC constraints.

Compared with REF, US10 does not see greater electricity production until 2035, while US50 promotes greater electrification from 2020 (Fig. 3a). In addition, more stringent PMMC reductions also expedite the retirement of high-emission-intensity sources in end-use sectors. For example, US10 does not have additional reductions in building biomass until 2035 relative to REF, while US50 reduces building biomass immediately in 2020 (Fig. 3b). More stringent PMMC reductions also lower the average PM$_{2.5}$ emission intensity for the same source (Fig. 3c). For example, the 2050 average PM$_{2.5}$ emission intensity for building biomass in US50 is 48% lower than that in REF, reflecting the greater efficiency and lower emission intensities of new building biomass units[28]. This result suggests that when phasing out high-emission-intensity sources in end-use sectors, the most cost-effective approach is to start with the older vintages.

**Regional PMMC reductions and actions**. The PMMC of state-level emissions depend in part on the magnitude and species of pollutant emissions and the population exposure within or near the emitting state. The distribution of PMMC in 2015 is mainly affected by the distribution of traditional polluting sources (such as coal-fired electric utilities) and population density, with the highest PMMC in OH and PA (Fig. 4a). In REF at 2050, PMMC declines in the East North Central states, including OH, and in

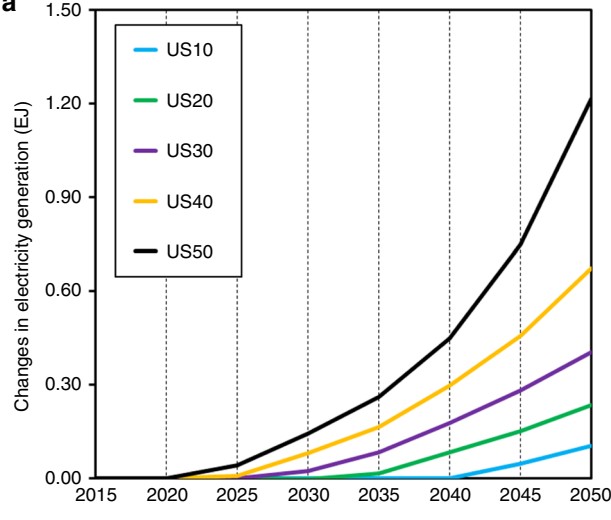

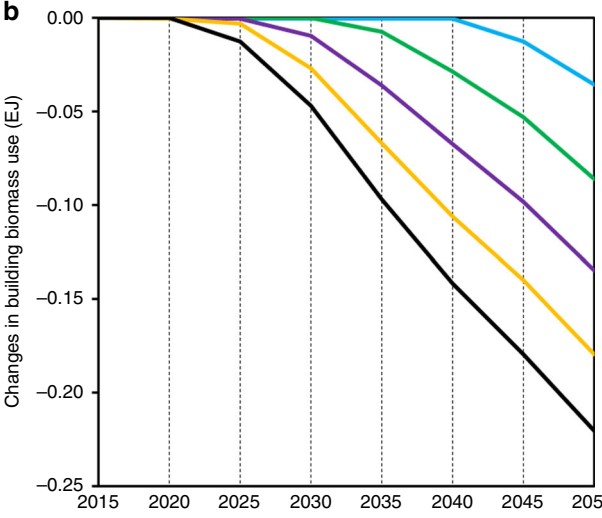

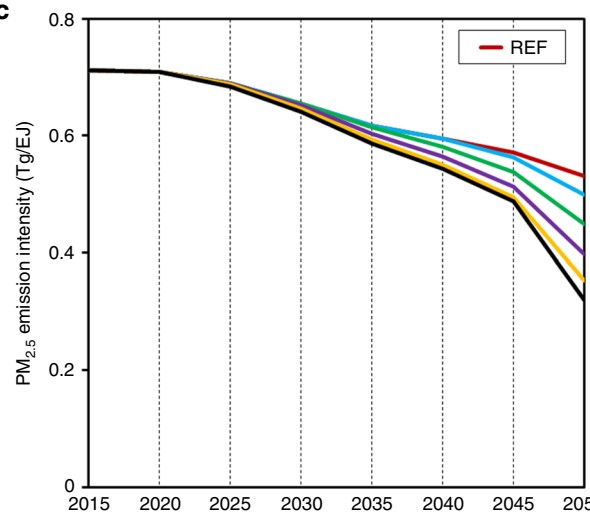

**Fig. 3 Effects of PM$_{2.5}$ reduction stringency on technology replacement.** Changes relative to REF for each modeling year for electricity generation (EJ) (**a**) and building biomass use (EJ) (**b**), and average PM$_{2.5}$ emission intensity (Tg EJ$^{-1}$ energy input) of building biomass (**c**). More stringent PM$_{2.5}$-related mortality costs (PMMC) reductions expedite electrification and the retirement of high-emission-intensity sources in end-use sectors. Emission intensity is also found to decrease for other major polluting sources, such as industrial coal use (Supplementary Fig. 10). Source data are provided as a Source Data file.

and Middle Atlantic states (Fig. 4c, d; Supplementary Fig. 11), which tend to have greater populations and larger emissions. Specifically, OH, PA, NY, MI, and WI have the greatest PMMC reductions in 2050 under both US30 and US50, suggesting lower costs for these states to replace high-emission-intensity sources with clean energy. All states have lower PMMC compared with REF, despite the challenges from increasing population. However, the reduction fraction varies widely across states. For example, under US50, the PMMC in MT and ND decreases by <5%, while WI and VT reduce their PMMC by >80% (Supplementary Fig. 12). The relative reductions in 2050 for some highly populated states such as TX and CA are also much less than the national target of 50% under US50.

The spatial pattern of PMMC reductions in Fig. 4 can be further explained by considering the state-level distribution of high-emission-intensity sources, particularly industrial coal, building sector biomass, and industrial liquids (Fig. 5). PA and OH together account for 31% of the national total industrial coal use in 2015, while growth of industrial coal is a major contributor to the increased PMMC at 2050 in most southeastern states[21]. Building biomass is broadly used in the US, with the highest absolute utilization in CA, which also has the largest reduction of building biomass use in US50 at 2050. US50 prevents the increased use of building biomass in the New England states seen in 2050 under REF, which is a major contributor to regional PMMC in the future. Industrial liquids are predominantly used in TX in 2015, accounting for 32% of the national total usage. TX also has the highest growth in industrial liquids use to 2050, together with slight increases among many southeastern and western states. Under US50, TX has the largest decrease in industrial liquids compared with REF in 2050, together with broad reductions in industrial liquid use for the eastern states and the west coast. Reducing these high-emission-intensity sources is accompanied by greater electricity generation in the states where those sources are located (Supplementary Fig. 13). The energy consumption changes for high-emission-intensity sources in 2030 have a similar geographic pattern for US30 and US50, although the magnitudes are smaller than the changes in 2050 (Supplementary Fig. 14).

**Cost-effectiveness of PMMC reductions.** From US10 to US50, the marginal benefits in 2050 for every 10% PMMC reduction are the same (~$33 billion, Fig. 6a), as defined by the PMMC reduction targets, while the estimated marginal costs increase by a factor of 50, from $0.3 billion in REF-US10 to $14.7 billion in US40–US50, as the most cost-effective control options are exhausted first (Fig. 6b). For all PMMC reduction scenarios considered, including US50, the benefits of reduced PMMC exceed the control costs. Even in US50, the marginal PM benefit of avoided mortality in 2050 is still over twice the marginal policy cost, while the national total health benefit in US50 is nearly seven times the corresponding policy cost.

US50 is also cost-effective for all states (Fig. 6c). In general, it is more cost-effective to reduce PMMC among the Cross-State Air

the Middle Atlantic states, including PA (Fig. 4b). However, despite an overall national reduction, PMMC are higher in 2050 for a number of Southern states (such as TX and FL), largely driven by population growth and increasing contributions from historically less-controlled sources in the industry sector[21].

When national PMMC reduction targets are applied, the greatest PMMC reductions are obtained in the East North Central

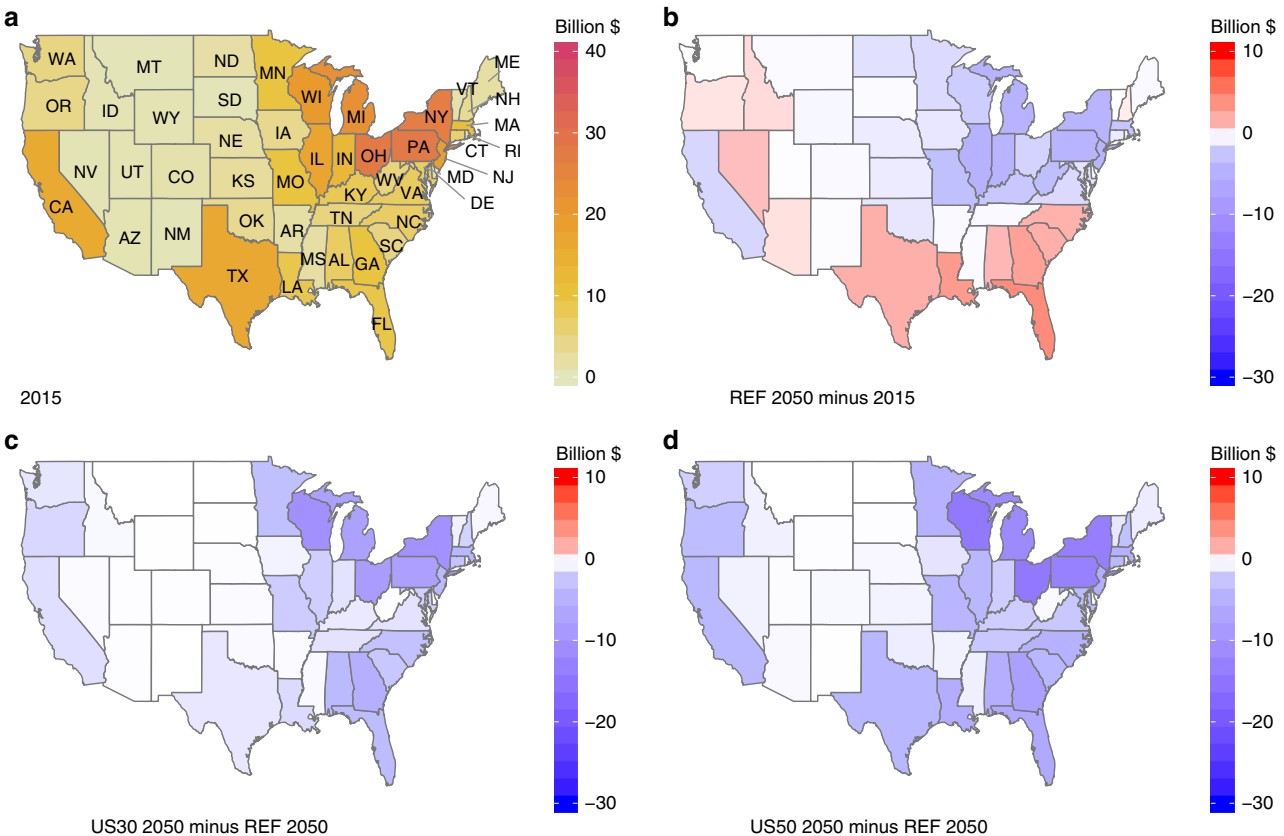

**Fig. 4 Regional PM$_{2.5}$-related mortality costs (PMMC) reductions.** Estimates of state-level PM$_{2.5}$ mortality costs (2018$ billion) in REF for 2015 (**a**), and absolute changes ($ billion) in REF by 2050 (**b**). Changes in PMMC relative to REF for US30 in 2050 (**c**) and US50 in 2050 (**d**). In REF, a number of states will face challenges in reducing their PMMC from 2015 to 2050. When applying national reduction targets, all states will have lower PMMC relative to 2015. This is meaningful for long-term air quality management, especially among states with high existing PMMC (such as PA and OH) and states with increasing PMMC in REF (such as TX and FL). Source data are provided as a Source Data file.

Pollution Rule (CSAPR) states (simple linear regression coefficient, $\beta = 7.5$, $N = 23$) compared with non-CSAPR states ($\beta = 5.1$, $N = 26$). CSAPR states are those identified in the CSAPR as having emissions that significantly contribute to downwind air pollution[29]. Note that the state-level PMMC coefficients represent the mortality impact of emissions from that state on both the population within that state and downwind of that state. The 23 CSAPR states together account for 60% of the total US population in 2015, while under US50 they account for 76% of the total PM health benefits and 71% of the total policy cost in 2050. Emission reductions in OH lead to the highest health benefits of $14.5 billion in 2050, as well as the highest policy cost of $2.2 billion. PMMC reductions among CSAPR states are achieved through emission reductions beyond the electric sector NO$_x$ and SO$_2$ emission caps that are mandated under CSAPR, which are fully phased in by 2030 under REF. State-level policy costs and health benefits of US50 and HR US50 are reported in Supplementary Tables 5 and 6, respectively. Notably, HR US50 is also cost-effective for all states.

## Discussion

Current US air pollution control is driven by compliance with air pollutant concentration standards and achieved through emission reductions from individual sources. Here we demonstrate a health-driven approach, in which PM$_{2.5}$-related mortality costs are directly constrained in the future. We simulate how national PMMC reduction targets could be achieved within a global human–earth systems model with US state-level energy system

representations, allowing the model to identify the states and actions that can most cost-effectively decrease future PMMC.

Our state-level results demonstrate the importance of devising state-specific strategies for improving air quality and protecting human health. Here, state-level actions are implicitly coordinated to represent the most cost-effective pathway to reach the specified national PMMC mitigation target. In this context, both the cost-effectiveness and the PMMC per capita can be very different across states (Supplementary Tables 5–7), and the relative reductions for some states can be much less than the national targets. Therefore, this approach may not achieve environmental equity across states, and may not be consistent with the goals of individual states. An extension of this work could explore cost-effective actions to achieve greater PMMC reductions in each state driven by state-specific targets.

We find limited co-benefits of CO$_2$ emission reductions under even stringent PMMC reductions up to US50. Similar findings are also shown in China's Clean Air Action, which successfully reduced major air pollutants from 2010 to 2017, but as accompanied by 16% increases in CO$_2$ emissions[30]. Another study found that wealthier cities in North America have reduced their PM$_{2.5}$ concentrations and mortality burdens considerably in recent years, while their CO$_2$ emissions have not declined in parallel[31]. This largely one-way co-benefit pathway is because of differences in the major actions for controlling CO$_2$ versus other pollutant emissions[16]. The most cost-effective CO$_2$ controls involve fuel switching or efficiency improvements[32,33], both of which also reduce air pollutants. In contrast, the most cost-effective air pollutant control strategies in the US do not

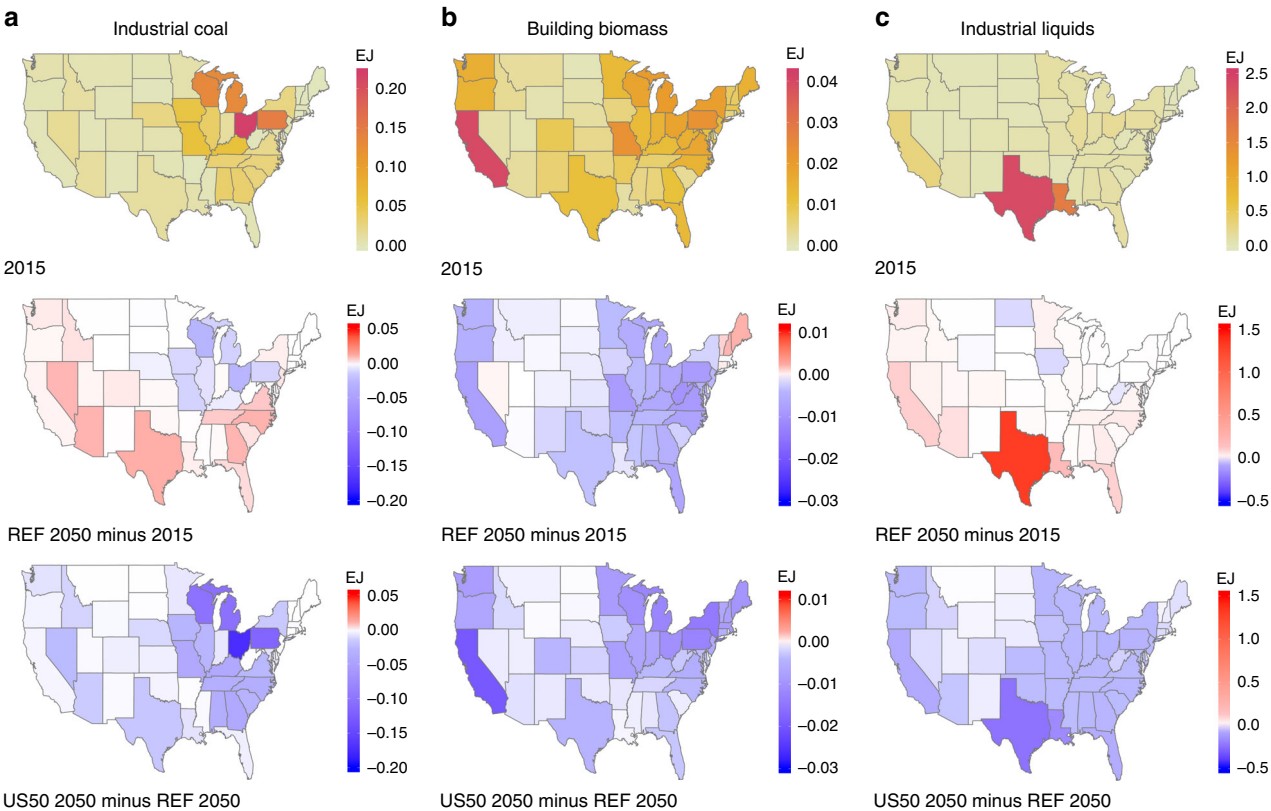

**Fig. 5 Regional actions of reducing PMMC.** The first row shows 2015 estimates of energy consumption in three high-emission-intensity sources—industrial coal (**a**), building biomass (**b**), and industrial liquids (**c**) for continental US states. The second row shows energy consumption changes for high-emission-intensity sources in 2050 compared with 2015, and the third row shows energy consumption changes in US50 relative to REF in 2050, as actions of reducing PMMC. Blue indicates reduced energy use in high-emission-intensity sources, and red indicates increased energy use in high-emission-intensity sources. Source data are provided as a Source Data file.

necessarily cause major changes in the general energy structure (Supplementary Figs. 7 and 8). Rather, they impose end-of-pipe controls or replace high-emission-intensity sources only, which have a limited effect on $CO_2$ emissions. Here, we find that most PMMC reductions can be achieved by reducing only a small portion of high-emission-intensity fuel use including industrial coal, industrial liquids, and building biomass, but the overall fuel mix does not change dramatically.

This work produces novel insights into potential architectures for multi-pollutant control strategies that are explicitly driven by PMMC reduction targets. Since GCAM-USA is a global modeling framework, the general approach demonstrated here can be extended with health impact representations for other countries or regions. Considering the differences in energy structures, domestic policy and geopolitical issues among different regions, the specific solutions may be very different. For example, China can obtain substantial air quality health benefits from large-scale fleet electrification, especially in populous megacities[34]. India's ambient air quality standard is achievable by mitigating household emission sources and rural electrification[35]. This methodology can also be extended with additional environmental and health endpoints, supporting a wide range of single and multi-objective outcome-driven policy analyses.

## Methods
**GCAM-USA.** The Global Change Assessment Model (GCAM) (http://jgcri.github.io/gcam-doc/) is a global human–earth system model designed for long-term, integrated assessment of the economy, energy, agriculture, land-use, water, and climate[36,37]. GCAM-USA builds upon GCAM by disaggregating the USA region to represent state-level energy supply and demand markets, nested within a global framework with 32 regions[38–40]. State-level economic activities are driven by exogenous assumptions regarding population, GDP, and labor productivity, which determine energy and service demands for each state. GCAM-USA simulates the operation of markets for energy goods and services from 2010 to 2100 in 5-year increments. Supply and demand are balanced in each modeling period by solving for a set of equilibrium prices for all agriculture, energy, emission, and policy-related markets. As a result, model solutions represent the most cost-effective combinations of technologies satisfying the energy and service demands for the current modeling period. Market shares of competing technologies and fuels are determined via logit functions, apportioning market share as a function of cost-effectiveness. Note that the resulting choices of technologies over time are not necessarily the economically optimal solution across the entire modeling period, because the dynamic recursive approach in GCAM simulates how the system evolves under imperfect foresight, in contrast to intertemporal optimization models. GCAM-USA has been employed for many US-specific policy analyses, addressing environmental implications of low-carbon pathways[22,39,41] and water use for power generation[42].

GCAM-USA estimates air pollutant emissions as the product of an economic activity (energy input or service output of a specific technology) and the corresponding emission coefficient. In our previous work[21,22,24], we have attempted to harmonize emission coefficients with EPA's National Emission Inventory (NEI) and its projections, and have included representations of important existing national and regional air quality, climate, and energy policies. These state-level updates allow a more a realistic reference scenario that serves as the baseline for examining additional $PM_{2.5}$ mortality cost reductions. Supplementary Tables 1 and 2 briefly summarize how energy use and air pollutant controls are modified in GCAM-USA for this study, and Supplementary Tables 8–10 include $PM_{2.5}$, $NO_x$, and $SO_2$ emission factors from 2015 to 2050, applied for all states. More detailed documentation and evaluation of GCAM-USA emissions can be found in our previous work[24].

**Marginal abatement curves (MACs) in GCAM-USA.** GCAM-USA includes marginal abatement curves (MACs) for electric sector $NO_x$ and $SO_2$ for existing coal-fired plants[24,43]. For existing power plants, the selective catalytic reduction (SCR) is the major control measure for $NO_x$, and the flue gas desulfurization (FGD) is the major control measure for $SO_2$. The costs of adding these controls are the same for all states (Supplementary Tables 11 and 12). In GCAM-USA, MACs

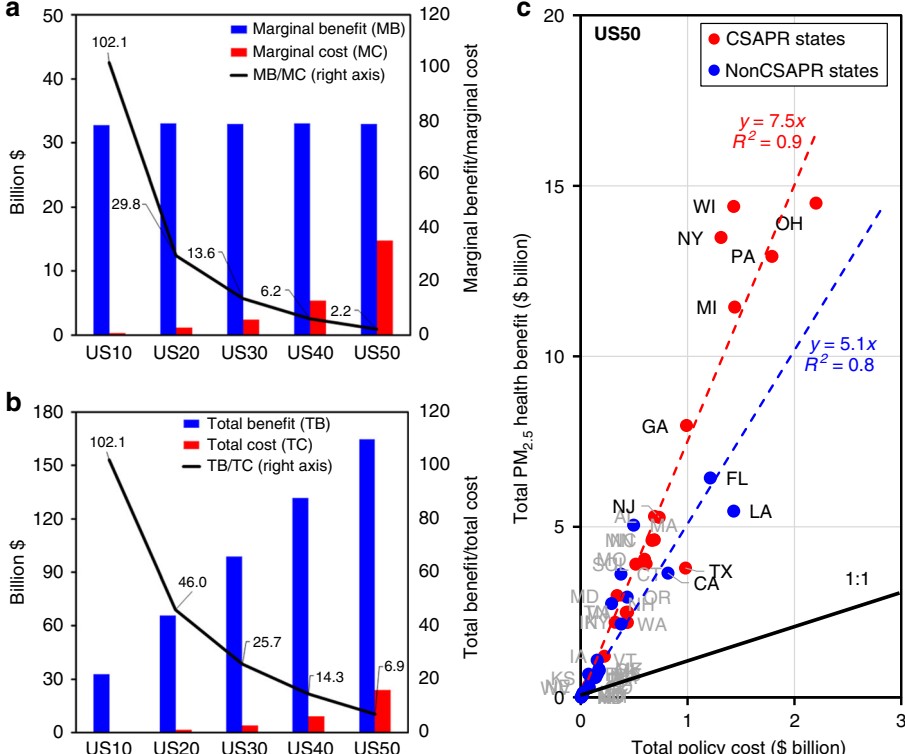

**Fig. 6 Cost-effectiveness of PM$_{2.5}$ mortality cost reductions.** Marginal PM$_{2.5}$ health benefit and marginal policy cost (**a**), total PM$_{2.5}$ health benefit and total policy cost (**b**) under PM$_{2.5}$ mortality cost reductions in 2050, and state-level estimates of total PM$_{2.5}$ health benefit and policy cost in US50 in 2050 (**c**). Source data are provided as a Source Data file.

can only be triggered when the corresponding emission markets are set, so we have applied MACs for both pollutants and their corresponding PMMCs. For CSAPR states, the emission markets are set by the emission caps of NO$_x$ and SO$_2$ for existing coal-fired power plants. This flexibility of adding additional emission controls through the MAC partially explains why CSAPR states can most cost-effectively reduce PMMC associated with their emissions.

In this work, we further develop MACs for NO$_x$, SO$_2$, and PM$_{2.5}$ emissions from coal, gas, and refined liquid sources in industrial sector. First, we examine existing industrial sources in the 2011 NEI to estimate the fraction of emissions to which controls could be applied, and then we combine these fractions with the corresponding control cost derived from the Control Strategy Tool[44]. For sources with different control options, an average control cost is applied to represent a one-step control. Supplementary Tables 13–21 show the resulting MACs of PM$_{2.5}$, NO$_x$, and SO$_2$ for industrial coal, gas, and refined liquid sources. Note that MACs are only applied to controllable sources. For states that do not have certain emissions from some source categories, or are already fully controlled, MACs are not available. These MACs allow electric and industry sectors to respond to policy targets by explicitly considering the application of end-of-pipe controls in addition to fuel switching and adopting more efficient technologies (Supplementary Table 22).

Although GCAM-USA is technology-rich in most sectors, the current version of GCAM-USA does not have technology-level representation of industrial fuel use, or explicit representation of high-efficiency wood stoves for residential heating. Adding these technological details in the future can more accurately quantify the technology turnover of high-polluting sources identified in this paper, as well as more explicit representation of technology-specific polices, such as the Maximum Achievable Control Technology (MACT) standard for industrial boiler[45] and state-specific residential wood stove emission standards.

**State-level PMMC in GCAM-USA.** The Estimating Air pollution Social Impact Using Regression (EASIUR) model[19,20] is a reduced-form model derived from regression of outputs of a chemical transport model. It has estimated the PM$_{2.5}$-attributable deaths within the US per tonne of inorganic air pollutant emissions (primary PM$_{2.5}$, SO$_2$, and NO$_x$) from each county in 2005. The concentration-response relation used for health impact estimation in EASIUR is derived from Krewski et al.[46], and \$8.6 M (in 2010 USD) is used for the value of a statistical life (VSL)[47].

These county-level death-per-tonne estimates were aggregated to the state level for the electricity, industry, transportation, and building sectors in 2005, based on the emission-weighted sum for each sector, and then adjusted for each future time

period modeled in GCAM-USA (2010 to 2050 in 5-year increments) to account for future changes in population exposure, baseline mortality rates, and VSL[19] (Supplementary Table 23). The state-level mortality costs presented here represent the impact of emissions from that state on the population within that state and all downwind states. State-level PMMC coefficients for all sectors vary widely across states (Supplementary Table 4). Our previous research[21,22] demonstrated that integrating PMMC coefficients into GCAM-USA provides an efficient and rapid approximation of PM$_{2.5}$ mortality impacts, allowing it to be used for evaluating large numbers of candidate management strategies to support decision-making.

The following caveats should be noted for the PMMC coefficients adopted in this study. First, PMMC coefficients consider only PM$_{2.5}$-related mortality impacts on adults from emissions of primary PM$_{2.5}$ and inorganic precursors, ignoring other emissions that lead to PM$_{2.5}$ pollution. Primary organic PM$_{2.5}$ is treated as inert in EASIUR, and secondary organic aerosol (SOA) is neglected because of the rapidly evolving understanding and representation of SOA formation in chemical transport models[20]. Anthropogenic emissions generally increase SOA through both volatile organic compounds (VOC) emissions and by enhancing the production of SOA from biogenic sources[48,49]. Therefore, the present analysis most likely underestimates the total mortality cost of PM$_{2.5}$ pollution, and omits the effects on SOA from other precursors such as NO$_x$ and effects of VOC emissions. The morbidity effects of PM$_{2.5}$ and the health effects of other pollutants such as ozone are not included, which would require the development of state-specific impact coefficients. For PM$_{2.5}$ mortality, EASIUR assumes a log-linear exposure-response function and equal toxicity of all PM$_{2.5}$, while nonlinearities may exist at low levels of exposure, and particles from different emission sources may have different toxicities[50,51]. A systematic quantification of these uncertainties in the chemical transport models used to develop EASIUR and similar reduced-form models is beyond the scope of this study.

Second, the use of PMMC coefficients assumes a linear relationship between changes in emissions and deaths. Although linearity may diminish for large changes in multiple pollutant emissions over a long-term[52,53], estimates of PMMC coefficients are generally considered valid within a factor of two, even over fairly large changes in baseline emissions[20]. Third, the PMMC coefficients adopted in this paper are derived based on 2005 emissions and meteorology[19], future work should seek to use PMMC coefficients determined specifically for future-year emissions and demographics, considering also potential changes in climate and consequent changes in PM$_{2.5}$[54]. Fourth, while we use state-level PMMC coefficients which allow us to capture projected large-scale population shifts within the US, finer scale shifts, such as changes in urbanization and aging, are also expected to occur[55]. The detailed process of incorporating EASIUR estimates into GCAM-USA and relevant discussion can be found in our previous work[21].

**Scenarios**. REF is developed to include air pollutant emission reductions and plausible energy changes from on the books regulations and policies including New Source Performance Standards[29], Tier 3 mobile vehicle fuel and emission standards[56], Cross-State Air Pollution Rule (CSAPR)[57] (Supplementary Tables 24 and 25), and the Regional Greenhouse Gas Initiative[58]. We also include the Corporate Average Vehicle Efficiency (CAFE) standards[59] (Supplementary Table 26), which affect air pollutant emissions through their impact on technology choices. Some state-specific requirements are considered. For example, we assume no new coal generation in California to reflect its SB-100 requirement[60]. We have discussed the role of these existing regulations in future pollutant emissions and PMMC in REF previously[22].

We consider two reference energy trajectories, BASE REF and HR REF. In BASE REF, key economic activity parameters such as electricity generation and projections of coal use in electric and industry sectors have been harmonized with the 2018 AEO[23] at national or regional levels[24].

In HR REF, capital costs of wind and solar power generation technologies are further modified so that the scenario results in greater renewable electricity penetration and more rapid electricity coal retirement (Supplementary Table 27). Alternative wind capital cost assumptions are developed using the Department of Energy 2016 Wind Technologies Market Report created by Lawrence Berkley National Laboratory (LBNL)[61]. Alternative solar PV capital cost assumptions are developed using the 2015 Utility-Scale Solar Report by LBNL[62]. To create an alternative capital cost trajectory, we averaged the lowest-cost projections across each wind and PV class, and calculated the percent change relative to the 2010 starting point[22]. In addition, we also modified the share weight assumptions of electricity coal use in GCAM-USA. Share weight is a calibration parameter used to reflect non-modeled factors that affect the competition among technologies. In most markets, share weights for 2010 are derived by comparing historic market penetrations versus model-predicted penetrations. For future years, the calibrated share weights transition incrementally to 1, which indicates no specific preference for any particular technology. To model technology shifting and state-specific fuel preference, share weights can be tuned based on expert judgement. In this work, the share weight of the conventional pulverized coal electricity production technology has been reduced from 0.35 to 0.03 in 2010, and kept constant in future modeling years to reflect a substantially reduced preference for coal-fired electric power plants in this scenario.

PMMC reduction targets (US10–US50) are assumed to be linearly applied beginning in 2020, in 5-year increments, to reach their designated percentage reduction (10–50%) relative to REF in 2050. These stylized scenarios are not intended to represent a realistic policy prescription, but are instead meant to show illustrative pathways that represent a continuation, to varying degrees, of the near-term decrease in mortality projected in REF and HR REF under current regulations.

**Representation of PMMC reduction targets in GCAM-USA**. Similar to pollutant emissions, PMMC are represented as byproducts associated with each economic activity in GCAM-USA. By setting reductions on the national PMMC, GCAM-USA creates a national PMMC market and places a shadow price on PMMC[63], which then impacts both technology choices and potential addition of end-of-pipe emission controls via the MACs. Existing and future technologies with emissions causing high $PM_{2.5}$ levels will be assigned a higher PMMC cost, providing an incentive to reduce PMMC by reducing use of these technologies, including the early retirement of existing equipment. The equilibrium price of PMMC for each modeling period reflects the iterative process of simultaneously solving all national and regional markets, including the US national PMMC market to meet the specified level of reduction.

**Cost-effectiveness of PMMC reductions**. We define the cost-effectiveness of a PMMC reduction scenario as the ratio of avoided mortality to the required control cost. This metric expands the traditional emission-impact sensitivity paradigm (avoided mortality per ton of emission reduction) to an energy-emission-impact framework (avoided mortality per dollar of control cost invested), illustrated in Eq. (1).

$$\frac{\Delta M}{\Delta C} = \frac{\Delta M}{\Delta E} \times \frac{\Delta E}{\Delta C}, \tag{1}$$

where $\Delta M$ represents the avoided mortality, $\Delta C$ represents the required control cost, and $\Delta E$ represents the avoided air pollutant emissions. Although many studies have separately investigated the $\Delta M/\Delta E$ component[53,55,64] and the $\Delta E/\Delta C$ component[65,66], the states and control actions that most cost-effectively reduce PMMC are identified in this study as those with the highest $\Delta M/\Delta C$.

The total $PM_{2.5}$ health benefit of a given PMMC reduction scenario in a given year is defined as the avoided PMMC relative to REF. The corresponding marginal $PM_{2.5}$ health benefit is estimated as the avoided PMMC relative to the last 10% PMMC reduction scenario, in a specific future year. The total policy cost of a given PMMC reduction scenario in a given year is estimated as the area below the marginal abatement curve (MAC) of PMMC[66,67] (Supplementary Fig. 15). The corresponding marginal policy cost is estimated as the difference between the policy costs of the current scenario and the next-lowest PMMC reduction scenario for the same modeling year.

The estimated policy cost in this paper represents the system-wide cost from all PMMC mitigation efforts, which is used as a general metric to better characterize the overall cost-effectiveness of PMMC reductions, neglecting feedback on the macroeconomy and population distribution. Evaluation of the cost-effectiveness of more detailed actions such as specific air pollutant control measures, energy efficiency and fuel switching require a bottom-up approach with detailed representation of technology-level controls[65,66].

**Reporting summary**. Further information on research design is available in the Nature Research Reporting Summary linked to this article.

## Data availability
Data used to perform this study can be found in the Supplementary Information and Supplementary Data. The source data underlying all figures are provided in the Source Data file. GCAM-USA is publicly available (https://github.com/JGCRI/gcam-core/releases). The additional GCAM-USA input files required to perform this study are available from the corresponding author upon request.

## Code availability
Code used in GCAM and R for this work is available from the corresponding author upon request.

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

## Acknowledgements
Yang Ou was supported by the Research Participation Program at the Center for Environmental Measurement and Modeling, U.S. Environmental Protection Agency, administered by the Oak Ridge Institute for Science and Education (ORISE). The views expressed in this article are those of the authors, and do not necessarily represent the views or policies of the U.S. Environmental Protection Agency.

## Author contributions
Y.O., J.J.W., D.L., and C.N. designed the study. Y.O. performed the research. S.J.S. provided guidance on GCAM modeling. Y.O. wrote the initial paper and all authors contributed to subsequent revisions.

## Competing interests
The authors declare no competing interests.
