## [Peer Review File · Nature Communications]

Reviewers' comments:

Reviewer #1 (Remarks to the Author):

I was asked to review the energy system aspect of this manuscript. The paper uses the GCAM-USA model which is a well-established and well-respected model in the energy modeling world, so there is not a concern from my perspective in relying on that model for this analysis. The energy system parameters used in GCAM-USA are generally aligned with EIA' 2018 Annual Energy Outlook (AEO). I assume the aligning was done against the reference scenario, but that was not specified. Aligning against AEO assumptions is common in the energy modeling community.

My primary concern with this paper is that only one future energy trajectory is presented, and that energy trajectory is not well documented. From what I can tell of that future energy trajectory, I do not believe it to be representative of what an actual future will look like (even a business as usual future). For example, Figure S1(a) shows that electricity generation from coal is flat over time, but given the many drivers of clean energy (e.g., the many 100% clean energy standards, the increased RPS requirements, etc.), this doesn't seem reasonable. As one example, I've plotted here the coal generation projections from two other organization to compare against the AEO 2018 reference case by EIA. These projections are the NREL 2018 Standard Scenarios Mid-Case and the BNEF 2018 New Energy Outlook. Both of these organization tend to have more bullish projections related to clean energy potential, so I am not advocating that your reference case show similar trends, but I do think that discussing the implications of your findings in the face of more of a clean energy transition is warranted. Would your finding still hold if the coal generation declines by half, with the associated PM reduction that would entail?

[Figure here included in attachment]

Relatedly, the limited change in the power sector evolution does give me pause as to what is represented in the model. Do you have the many renewable and clean energy standards in the model that I referenced above? If so, then I have fewer concerns about your ability to represent any transitions here. Or what about the CAFE standards? And California's SB-100 requirement? I didn't see mention of those. And from what I could tell, the cost and performance input assumptions for electricity technologies were Iyer et al. (2017) which used the AEO 2016 for input assumptions. For solar in particular, those projections are relatively old (the AEO 2016 reference case PV prices never get below \$1500/kW by 2040, but current prices are already at or below that point – see slide 18 at https://emp.lbl.gov/sites/default/files/lbnl_utility_scale_solar_2018_edition_slides.pdf), and using high renewable energy cost would lead to less of a transition away from coal.

As you can probably tell from the comments above, I am interested in more documentation about your assumptions. One in particular that I think should be reported in the PM emission rates from the various technology types you represent in the energy sector. I don't feel like anything is "wrong" with the modeling, only that some assumptions might be dated and might not be reflective of the transition that is already happening in the energy sector, so ensuring that the findings are robust against a transition would make this paper more meaningful to me.

Reviewer #2 (Remarks to the Author):

One of the consequences of Rachel Carson's Silent Spring was that air pollution control programs that had previously been part of the US Public Health Service were moved into the newly formed

Environmental Protection Agency. Air pollution control thus came to be driven not by measures of its effects on human health, but rather by a legal and regulatory framework that was focused on compliance with emissions standards and meeting pollution levels. This framework has had significant successes and has greatly reduced levels of air pollution in the United States, although more than 100,000 deaths are still ascribed each year in the United States to ambient air pollution.

The authors of this very interesting manuscript propose a new approach to modeling efficacy of air pollution control standards, an approach that is focused specifically on seeking the most cost-effective ways to achieve reductions in pollution related mortality costs. In a sense, this proposed approach brings air pollution control back to its beginnings in public health.

To assess the relative efficacy of various pollution control strategies, the authors examine reductions in pollution related mortality costs that could be achieved under a reference scenario, as well as under 5 alternate scenarios of increasing stringency. In the reference scenario, no additional future regulations are assumed, whereas each of the other scenarios apply increasingly strict national reduction targets focused on reduction in mortality costs.

The findings are very interesting. Most importantly, they find that very large reductions in pollution related mortality costs could be achieved by focusing on a small number of high emission intensity sources, such as industrial coal and building biomass. They acknowledge that this approach will do little to reduce CO₂ emissions or to slow the pace of climate change, but they emphasize that these "end-of-pipe" approaches will be highly effective in achieving short-term reductions in morbidity and mortality and thus in reducing the costs of pollution related disease and death.

Review #1

REMARKS TO THE AUTHOR

I was asked to review the energy system aspect of this manuscript. The paper uses the GCAM-USA model which is a well-established and well-respected model in the energy modeling world, so there is not a concern from my perspective in relying on that model for this analysis. The energy system parameters used in GCAM-USA are generally aligned with EIA' 2018 Annual Energy Outlook (AEO). I assume the aligning was done against the reference scenario, but that was not specified. Aligning against AEO assumptions is common in the energy modeling community.

My primary concern with this paper is that only one future energy trajectory is presented, and that energy trajectory is not well documented. From what I can tell of that future energy trajectory, I do not believe it to be representative of what an actual future will look like (even a business as usual future). For example, Figure S1(a) shows that electricity generation from coal is flat over time, but given the many drivers of clean energy (e.g., the many 100% clean energy standards, the increased RPS requirements, etc.), this doesn't seem reasonable. As one example, I've plotted here the coal generation projections from two other organization to compare against the AEO 2018 reference case by EIA. These projections are the NREL 2018 Standard Scenarios Mid-Case and the BNEF 2018 New Energy Outlook. Both of these organization tend to have more bullish projections related to clean energy potential, so I am not advocating that your reference case show similar trends, but I do think that discussing the implications of your findings in the face of more of a clean energy transition is warranted.

Would your finding still hold if the coal generation declines by half, with the associated PM reduction that would entail?

[Figure here included in attachment]

Response: We thank the reviewer for highlighting the need to discuss our harmonization with AEO and for the comments related to the use of AEO's generally conservative assumptions regarding renewables. In response, we now clarify our harmonization with AEO and have **added two additional simulations with greater renewable power generation.**

Our changes are summarized as below:

1) In page 4-5s, the main conclusion is modified to read “Our results suggest that even under a scenario with limited transitions from coal towards renewable energy, substantial health benefits can still be cost-effectively achieved by using electricity to replace sources with high primary PM_{2.5} emission intensities, including industrial coal, building biomass, and industrial liquids. This finding also holds under a scenario with a more aggressive transition towards renewable energy.”

2) In page 5, the first two paragraphs of the Results section are updated to introduce our two energy trajectories to read:

“We define a reference scenario (BASE REF) that includes updated baseline assumptions about technology trajectories harmonized with the 2018 Annual Energy Outlook, as well as an alternative reference scenario (HR REF) with much higher shares of wind and solar generation (e.g., 48% versus 25% in 2050), intended to reflect the recent and rapid growth in renewables. For both, major air pollutant and CO₂ emission regulations currently in place are represented²³, assuming no additional future regulations (Methods). Electric sector coal use in HR REF decreases by 51% from 2015 to 2050, compared to the 8% increase in BASE REF (Supplementary Figure 13).

Relative to BASE REF, five additional scenarios (US10-US50) apply increasingly stringent national PMMC reduction targets of 10% (US10) to 50% (US50) reductions in 2050 relative to BASE REF in 2050. Similarly, a 50% PMMC reduction target is applied to HR REF (HR US50). We focus on the BASE scenarios in this study because the coal and renewable assumptions under BASE REF pose greater challenges for reducing PMMC (Supplementary Methods, Section 5). For simplicity, REF and US10-US50 throughout the rest of the study refer to scenarios based on BASE REF. HR REF and HR US50 are used to evaluate the robustness of the major findings using BASE.”

3) In page 6, the following sentences in sub-section “Effects of 50% PMMC reduction” are updated to confirm that our findings in Fig. 1 are valid under both BASE and HR energy trajectories.

“For both BASE and HR scenarios, reducing PMMC by 50% leads to very little change in national total energy use in 2050, relative to REF (Fig. 1a, b).”

“Under the HR scenarios (Fig. 1c and Fig. 1f), the decrease in these high emission intensity sources is also equivalent to 2% of the total energy consumption in 2050 but still contributes to 70% of the PMMC reductions, with additional contributions from decreased electric coal use.”

4) In page 8, the following sentences is added.

“This finding is valid under both BASE and HR energy trajectories (Supplementary Figure 15).”

5) Fig. 1 is updated to show results of both energy trajectories. The original panels (c) and (f) are moved to SI. Fig.1 title is updated to read

“Figure 1 National energy system (a-c) and PM_{2.5} mortality cost (d-f) responses to the 50% PM_{2.5} mortality cost reduction (US50) based on two reference scenarios. National total energy use (EJ) by fuel (a), sectoral changes in energy use (EJ) by fuel for BASE US50 relative to BASE REF (b) and HR US50 relative to HR REF (c); national total PM_{2.5} mortality costs (billion 2018\$) by fuel (d), and sectoral changes in PM_{2.5} mortality costs (billion \$) by fuel for US50 reductions under BASE (e) and HR scenarios (f). With the exception of the bars labeled 2015, all values are at 2050. Changes in the transportation sector are negligible. The “Non-

polluting energy” category includes geothermal, hydropower, wind, and solar power in the electricity sector, and hydrogen and electricity in enduse sectors.”

6) In page 8, the following sentence is added to read “This finding is valid under both BASE and HR energy trajectories (Supplementary Figure 15).”

7) In page 18-19, the “Scenarios” sub-section in Methods, the following paragraph is added to explain two energy trajectories

“We consider two reference energy trajectories, BASE REF and HR REF. In BASE REF, key economic activity parameters such as electricity generation and projections of coal use in electric and industry sectors have been harmonized with the 2018 Annual Energy Outlook at national or regional levels²⁴ (Supplementary Methods, Section 1). In HR REF, capital costs of wind and solar power generation technologies are further modified such that the scenario projects greater renewable electricity penetration and more rapid electricity coal retirement (Supplementary Methods, Section 5).”

8) Starting from SI page 40, a new section “5 Alternative high renewables energy trajectory” is added to document assumptions and methods used to create the High renewables energy trajectory. The new section is as below, including two supplementary tables and three supplementary figures.

5 Alternative high renewables energy trajectory

As documented in Section 1, the BASE REF scenario is harmonized with the 2018 AEO projections (EIA 2018). Here we additionally construct an alternative REF scenario based on a “high renewable” energy trajectory (HR REF) to reflect greater renewable electricity penetration and more rapid retirement of coal-fired power plants. HR REF is constructed by the following two modifications:

(1) **Modified wind and solar PV capital cost assumptions in GCAM-USA** (Supplementary Table 26). Alternative wind capital cost assumptions are developed using the Department of Energy 2016 Wind Technologies Market Report, which was created by Lawrence Berkley National Laboratory (LBNL) (Wiser & Bolinger, 2017). Alternative solar PV capital cost assumptions are developed using the 2015 Utility-Scale Solar Report by LBNL (Bolinger & Seel, 2016). To create an alternative capital cost trajectory, we averaged the lowest-cost projections across each wind and PV class, then calculated the percent change relative to the 2010 starting point. A more systematic documentation of this approach can be found in the Supporting Information of Ou et al. (2018).

(2) **Modified shareweight assumptions of electricity coal use.** Shareweight is a calibration parameter in GCAM-USA that is used to reflect non-modeled factors that affect the competition among technologies. In most markets, shareweights for 2010 are derived by comparing historic market penetrations versus model-predicted penetrations. For future years, the calibrated shareweights transition incrementally to 1, which indicates no specific preference for any particular technology. To model technology shifting and state-specific fuel preference,

shareweights can be tuned based on expert judgement. In this work, the shareweight of the conventional pulverized coal electricity production technology has been reduced from 0.35 to 0.03 for all modeling periods to reflect a substantially reduced preference for coal-fired electric power plants in this scenario.

Supplementary Figure 13 shows electricity generation by technology for BASE REF and HR REF. HR REF has much higher shares of wind and solar generation in 2050 (48% versus 25%). Compared with an 8% increased coal use in the BASE REF, electricity coal use in the HR REF decreased by 51% from 2015 to 2050. The modified coal trajectory is consistent with the NREL 2018 Standard Scenarios Mid-Case (Cole et al. 2018). While Bloomberg (2016) projected even more aggressive coal retirement, the current HR REF is sufficient to evaluate whether alternative energy trajectory would result in different conclusions to our study. Projected emissions of major air pollutants are also considerably lower in HR REF (Supplementary Figure 15). In 2050, SO₂ emissions decrease by 47% and 21% from 2015 in HR REF and BASE REF, correspondingly. Primary PM_{2.5} emissions decrease by 65% and 27% from 2015 in HR REF and BASE REF, correspondingly. These additional pollutant emission reductions lead to a much lower baseline PMMC projection.

Similar to the main text, we applied a 50% PMMC reduction target to the HR REF, noted as HR US50. As seen in Figure 1, Supplementary Figure 14, Supplementary Figure 15, and Supplementary Table 27, the major findings under BASE energy trajectories shown in the main text are still valid under the HR energy trajectory: a significant amount of PMMC reductions can be achieved cost effectively by electrifying a small percentage of high emission intensity sources, including industrial coal, industrial liquids, and building biomass. For HR US50, electric coal use is further reduced for the following two reasons. First, wind and solar costs are much lower in HR US50 than BASE US50, making it more cost-effective to displace coal generation. Second, the HR US50 mortality reduction target, which is relative to the HR 2050 reference case value, is more stringent than the BASE US50 target because the HR REF already has a lower baseline PMMC compared to BASE REF (Figure 1).

Supplementary Table 26 Capital cost assumptions for electricity generation technologies (2018\$/kW).

Technologies ¹	BASE ²			HR ³		
	2015	2030	2050	2015	2030	2050
Wind	2433	1747	1737	1985	1513	1532
PV	2970	1051	929	1877	626	374

¹ All technologies not presented are assumed to be the same as GCAM-USA default values

² Adopted from Iyer et al.(2017) - advanced technology scenario cost assumptions

³ Alternative wind capital cost assumptions are developed from Wisler & Bolinger (2017), and alternative utility-scale solar PV capital cost assumptions are developed from Bolinger & Seel (2016)

Supplementary Figure 13 Electricity generation by technology (EJ) in BASE REF (a) and HR REF (b).

Supplementary Figure 14 **National air pollutant emissions (a-b) and PM_{2.5} mortality cost (c-d) responses to the 50% PM_{2.5} mortality cost reduction (US50) under two future energy trajectories.** Sectoral changes in emissions (Tg) by pollutant under BASE trajectory (a) and HR trajectory (b); sectoral changes in PM_{2.5} mortality costs (billion \$) by pollutant under BASE trajectory (c) and HR trajectory (d). Reductions are relative to the REF of the corresponding energy trajectory in 2050. Changes in the transportation sector are negligible.

Supplementary Figure 15 Effects of PM_{2.5} reduction stringency on national emissions of primary PM_{2.5} (a), SO₂ (b), NO_x (c), and CO₂ (d) in BASE and HR scenarios.

Supplementary Table 27 State-level estimates of marginal policy cost and PM_{2.5} health benefit in HR US50 in 2050 (Unit: 2018 \$ Billion), relative to HR REF.

	Marginal policy cost	Marginal health benefit		Marginal policy cost	Marginal health benefit
AL	1.58	1.77	NC	1.57	1.76
AR	0.52	0.58	ND	0.01	0.01
AZ	0.16	0.18	NE	0.32	0.36
CA	3.71	4.16	NH	2.09	2.35
CO	0.30	0.34	NJ	4.00	4.49
CT	2.89	3.24	NM	0.04	0.05
DC	0.10	0.11	NV	0.12	0.14
DE	0.40	0.45	NY	8.30	9.31
FL	4.92	5.52	OH	8.80	9.87
GA	2.57	2.88	OK	0.80	0.90
IA	0.86	0.96	OR	2.04	2.29
ID	0.23	0.26	PA	7.35	8.25
IL	3.24	3.64	RI	0.56	0.63
IN	2.65	2.97	SC	1.05	1.18
KS	0.59	0.66	SD	0.07	0.08
KY	1.58	1.77	TN	0.83	0.93
LA	5.88	6.59	TX	4.29	4.81
MA	3.77	4.24	UT	0.14	0.16
MD	1.23	1.38	VA	1.29	1.45
ME	0.77	0.87	VT	0.87	0.97
MI	6.82	7.65	WA	1.63	1.83
MN	3.88	4.35	WI	4.80	5.38
MO	1.55	1.74	WV	0.35	0.40
MS	0.55	0.61	WY	0.05	0.05
MT	0.01	0.01	National total	102.12	114.58

*Both marginal policy cost and marginal health benefit in HR US50 are estimated directly relative to HR REF because HR US10 - US40 are not modelled. In this case, the marginal abatement curve is approximated to linear, so the reported marginal policy costs are overestimated.

Relatedly, the limited change in the power sector evolution does give me pause as to what is represented in the model. Do you have the many renewable and clean energy standards in the model that I referenced above? If so, then I have fewer concerns about your ability to represent any transitions here. Or what about the CAFE standards? And California’s SB-100 requirement? I didn’t see mention of those. And from what I could tell, the cost and performance input assumptions for electricity technologies were Iyer et al. (2017) which used the AEO 2016 for input assumptions. For solar in particular, those projections are relatively old (the AEO 2016 reference case PV prices never get below \$1500/kW by 2040, but current prices are already at or below that point – see slide 18 at https://emp.lbl.gov/sites/default/files/lbnl_utility_scale_solar_2018_edition_slides.pdf), and using high renewable energy cost would lead to less of a transition away from coal.

Response:

1) **Policy representation.** One strength of our modeling approach is the ability to incorporate US state-level emission and energy regulations, capturing more realistic emission trajectories. Our previous work (references 21-23 in the main text) have documented these policies and their implications on future air quality management. To make it clear, we made the following change.

In page 18, the first paragraph of “Scenarios” is updated to read “REF is developed to include air pollutant emission reductions and plausible energy changes from “on the books” regulations and policies including New Source Performance Standards²⁹, Tier 3 mobile vehicle fuel and emission standards⁵⁵, Cross-State Air Pollution Rule (CSAPR)⁵⁷, and the Regional Greenhouse Gas Initiative⁵⁸ (Supplementary Methods, Section 1). We also include the Corporate Average Vehicle Efficiency standards⁵⁹, which affect air pollutant emissions through their impact on technology choices. Some state-specific requirements are considered. For example, we assume no new coal generation in California to reflect its SB-100 requirement⁶⁰. We have discussed the role of these existing regulations in future pollutant emissions and PMMC in REF previously²¹⁻²³.” In addition, our original Supplementary Table 1 and 2 also listed our modifications and justifications, most of which are aimed at reflecting various policies and market trends.

2) **Technology cost.** When constructing the “HR” energy trajectory, we have further updated the capital cost assumptions of wind and solar technologies with greater reductions in future. Specific cost assumptions for BASE and HR energy trajectories are shown in Supplementary Table 26 as below. The detailed method is documented in the newly added Section 5 in SI (see our response to the previous question above).

Supplementary Table 26 Capital cost assumptions for electricity generation technologies (2018\$/kW).

Technologies ¹	BASE ²			HR ³		
	2015	2030	2050	2015	2030	2050
Wind	2433	1747	1737	1985	1513	1532
PV	2970	1051	929	1877	626	374

¹ All technologies not presented are assumed to be the same as GCAM-USA default values

² Adopted from Iyer et al.(2017) - advanced technology scenario cost assumptions

³ Updated wind capital cost assumptions are developed from Wiser & Bolinger (2017), and updated utility-scale solar PV capital cost assumptions are developed from Bolinger & Seel (2016)

As you can probably tell from the comments above, I am interested in more documentation about your assumptions. One in particular that I think should be reported in the PM emission rates from the various technology types you represent in the energy sector. I don't feel like anything is "wrong" with the modeling, only that some assumptions might be dated and might not be reflective of the transition that is already happening in the energy sector, so ensuring that the findings are robust against a transition would make this paper more meaningful to me.

Response: As suggested, we have included Supplementary Tables 3-5 to show emission factors for primary PM_{2.5}, NO_x, and SO₂ as below.

Supplementary Table 3 Future-year (2015 to 2050) emission factors for electric sector sources in GCAM-USA (Tg/EJ energy input). These emission factors represent New Source Performance Standards (NSPS) requirements.

Technologies	NO _x	SO ₂	PM _{2.5}
Coal (conventional pulverized)	3.0E-02	2.6E-02	Calibrated ¹
Coal (conventional pulverized with CCS)	3.0E-02	2.6E-02	1.6E-02
Coal (IGCC)	5.6E-03	0.0E+00	3.2E-03
Coal (IGCC CCS)	5.6E-03	0.0E+00	3.2E-03
Gas (CC)	4.7E-03	0.0E+00	1.3E-04
Gas (CC CCS)	4.7E-03	0.0E+00	1.3E-04
Gas (steam/CT)	4.7E-03	0.0E+00	3.4E-03
Refined liquids (CC)	2.4E-01	3.1E-02	6.2E-03
Refined liquids (CC CCS)	2.4E-01	3.1E-02	6.2E-03
Refined liquids (steam/CT)	2.4E-01	3.1E-02	6.2E-03
Biomass (conventional)	8.6E-03	3.4E-02	3.1E-02
Biomass (conventional with CCS)	8.6E-03	3.4E-02	3.1E-02
Biomass (IGCC)	4.7E-03	0.0E+00	1.3E-04
Biomass (IGCC with CCS)	4.7E-03	0.0E+00	1.3E-04

¹ Primary PM_{2.5} EFs of conventional pulverized coal plants for future years are the same as the calibrated values in 2010, differing by state

Supplementary Table 4 Future-year (2015 to 2050) emission factors for industrial sector sources in GCAM-USA (Tg/EJ energy input)

Industrial fuel use		NO_x	SO₂	PM_{2.5}
coal/coal cogeneration	2015	1.2E-01	5.2E-01	2.4E-03
	2020	1.2E-01	2.1E-01	2.4E-03
	2025	1.2E-01	2.1E-01	2.4E-03
	2030	1.2E-01	2.1E-01	2.4E-03
	2035	1.2E-01	2.1E-01	2.4E-03
	2040	1.2E-01	2.1E-01	2.4E-03
	2045	1.2E-01	2.1E-01	2.4E-03
	2050	1.2E-01	2.1E-01	2.4E-03
gas/gas cogeneration	2015	2.3E-02	7.1E-04	8.9E-03
	2020	2.6E-02	6.9E-04	8.8E-03
	2025	2.7E-02	6.9E-04	8.9E-03
	2030	2.9E-02	7.3E-04	9.3E-03
	2035	3.1E-02	7.5E-04	9.4E-03
	2040	3.2E-02	7.7E-04	9.6E-03
	2045	3.2E-02	7.7E-04	9.7E-03
	2050	3.3E-02	7.7E-04	9.6E-03
liquid fuels/liquid fuels cogeneration	2015	7.7E-02	2.7E-02	4.3E-02
	2020	6.3E-02	2.1E-02	3.9E-02
	2025	6.3E-02	2.1E-02	4.0E-02
	2030	6.5E-02	2.2E-02	4.8E-02
	2035	6.9E-02	2.3E-02	4.5E-02
	2040	7.5E-02	2.5E-02	4.8E-02
	2045	7.2E-02	2.5E-02	4.7E-02
	2050	7.5E-02	2.6E-02	5.0E-02
biomass/ biomass cogeneration	2015	2.3E-02	1.3E-03	5.0E-03
	2020	2.2E-02	1.3E-03	4.3E-03
	2025	2.0E-02	1.3E-03	3.7E-03
	2030	1.9E-02	1.3E-03	3.4E-03
	2035	1.8E-02	1.3E-03	3.1E-03
	2040	1.7E-02	1.3E-03	3.0E-03
	2045	1.6E-02	1.3E-03	2.7E-03
	2050	1.6E-02	1.3E-03	2.6E-03

Supplementary Table 5 Emission factors for building sector sources in GCAM-USA (Tg/EJ energy input).

	Technology	NO _x	SO ₂	PM _{2.5}	
Commercial	gas range	5.3E-02	3.5E-03	6.7E-03	
	gas range hi-eff	5.3E-02	3.5E-03	6.7E-03	
	gas cooling	5.3E-02	3.5E-03	6.7E-03	
	coal furnace	2.5E-01	8.6E-01	2.0E-02	
	fuel furnace	3.0E-02	7.6E-03	3.4E-03	
	gas furnace	5.3E-02	3.5E-03	6.7E-03	
	gas furnace hi-eff	5.3E-02	3.5E-03	6.7E-03	
	wood furnace	4.5E-02	5.2E-03	9.8E-02	
	fuel water heater	3.0E-02	7.6E-03	3.4E-03	
	gas water heater	5.3E-02	3.5E-03	6.7E-03	
	gas water heater hi-eff	5.3E-02	3.5E-03	6.7E-03	
	gas	5.3E-02	3.5E-03	6.7E-03	
	refined liquids	3.0E-02	7.6E-03	3.4E-03	
	Residential	clothes dryer	3.7E-02	4.6E-04	3.4E-03
		gas range	3.7E-02	4.6E-04	3.4E-03
		gas range hi-eff	3.7E-02	4.6E-04	3.4E-03
LPG range		1.2E-02	2.5E-02	1.6E-03	
LPG range hi-eff		1.2E-02	2.5E-02	1.6E-03	
fuel furnace		1.2E-02	2.5E-02	1.6E-03	
fuel furnace hi-eff		1.2E-02	2.5E-02	1.6E-03	
gas furnace		3.7E-02	4.6E-04	3.4E-03	
gas furnace hi-eff		3.7E-02	4.6E-04	3.4E-03	
fuel water heater		1.2E-02	2.5E-02	1.6E-03	
fuel water heater hi-eff		1.2E-02	2.5E-02	1.6E-03	
gas water heater		3.7E-02	4.6E-04	3.4E-03	
gas water heater hi-eff		3.7E-02	4.6E-04	3.4E-03	
gas appliances		3.7E-02	4.6E-04	3.4E-03	
refined liquids		1.2E-02	2.5E-02	1.6E-03	
wood furnace - 2010		6.9E-02	9.3E-03	5.6E-01	
wood furnace - 2015		6.9E-02	9.3E-03	5.1E-01	
wood furnace - 2020		6.9E-02	9.3E-03	4.7E-01	
wood furnace - 2025	6.9E-02	9.3E-03	4.2E-01		
wood furnace - 2030	6.9E-02	9.3E-03	3.7E-01		

* Only residential wood furnace have year-specific EFs for PM_{2.5} emissions

Review #2

REMARKS TO THE AUTHOR

One of the consequences of Rachel Carson's Silent Spring was that air pollution control programs that had previously been part of the US Public Health Service were moved into the newly formed Environmental Protection Agency. Air pollution control thus came to be driven not by measures of its effects on human health, but rather by a legal and regulatory framework that was focused on compliance with emissions standards and meeting pollution levels. This framework has had significant successes and has greatly reduced levels of air pollution in the United States, although more than 100,000 deaths are still ascribed each year in the United States to ambient air pollution. The authors of this very interesting manuscript propose a new approach to modeling efficacy of air pollution control standards, an approach that is focused specifically on seeking the most cost-effective ways to achieve reductions in pollution related mortality costs. In a sense, this proposed approach brings air pollution control back to its beginnings in public health.

To assess the relative efficacy of various pollution control strategies, the authors examine reductions in pollution related mortality costs that could be achieved under a reference scenario, as well as under 5 alternate scenarios of increasing stringency. In the reference scenario, no additional future regulations are assumed, whereas each of the other scenarios apply increasingly strict national reduction targets focused on reduction in mortality costs.

The findings are very interesting. Most importantly, they find that very large reductions in pollution related mortality costs could be achieved by focusing on a small number of high emission intensity sources, such as industrial coal and building biomass. They acknowledge that this approach will do little to reduce CO₂ emissions or to slow the pace of climate change, but they emphasize that these "end-of-pipe" approaches will be highly effective in achieving short-term reductions in morbidity and mortality and thus in reducing the costs of pollution related disease and death.

Response: We thank the reviewer for the insightful comments and context. While there is no specific requirement by the reviewer, we have updated our introduction and discussion sections to incorporate the reviewer's remarks. Specifically,

In page 3, the following two sentences are added at the beginning of the first and second paragraphs, respectively.

"Air pollution control in the US is focused on compliance with emissions limits and meeting air quality standards."

"A primary goal of air pollution control is to protect human health."

In page 12, the following sentence is added at the beginning of our discussion section to read: "Current US air pollution control is driven by compliance with air pollutant concentration standards, achieved through emission reductions from individual sources. Here we demonstrate a health-driven approach in which PM_{2.5}-related mortality costs are directly constrained in the future. We simulate how national PPMC reduction targets could be achieved within a global

human-earth systems model with US state-level energy system representations, allowing the model to identify the states and actions that can most cost-effectively decrease future PMMC.”

REVIEWERS' COMMENTS:

Reviewer #1 (Remarks to the Author):

The authors have done considerable work to update the analysis and manuscript in response to my original comments. I thank the authors for their effort, and feel like the manuscript is much improved and more relevant to the current happenings in the power sector.

From the stand-point of the power sector representation in this paper, I feel like things are robust and represented well. I have no concerns with this paper proceeding forwarding.

Review #1

REMARKS TO THE AUTHOR

The authors have done considerable work to update the analysis and manuscript in response to my original comments. I thank the authors for their effort, and feel like the manuscript is much improved and more relevant to the current happenings in the power sector.

From the stand-point of the power sector representation in this paper, I feel like things are robust and represented well. I have no concerns with this paper proceeding forwarding.

Response: We are appreciative of the reviewer's comments and agree that the changes that were made to address his or her earlier comments result in a more robust study.